# Impact of Surface Trap States on Electron and Energy Transfer in CdSe Quantum Dots Studied by Femtosecond Transient Absorption Spectroscopy

**DOI:** 10.3390/nano14010034

**Published:** 2023-12-22

**Authors:** Hongbin Dou, Chunze Yuan, Ruixue Zhu, Lin Li, Jihao Zhang, Tsu-Chien Weng

**Affiliations:** 1School of Physical Science and Technology, ShanghaiTech University, Shanghai 201210, China; douhb@shanghaitech.edu.cn (H.D.); lilin1@shanghaitech.edu.cn (L.L.); zhangjh5@shanghaitech.edu.cn (J.Z.); 2Center for Transformative Science, ShanghaiTech University, Shanghai 201210, China

**Keywords:** quantum dots, surface trap states, transient absorption, photoelectron transfer, triplet-triplet energy transfer

## Abstract

The presence of surface trap states (STSs) is one of the key factors to affect the electronic and optical properties of quantum dots (QDs), however, the exact mechanism of how STSs influence QDs remains unclear. Herein, we demonstrated the impact of STSs on electron transfer in CdSe QDs and triplet-triplet energy transfer (TTET) from CdSe to surface acceptor using femtosecond transient absorption spectroscopy. Three types of colloidal CdSe QDs, each containing various degrees of STSs as evidenced by photoluminescence and X-ray photoelectron spectroscopy, were employed. Time-resolved emission and transient absorption spectra revealed that STSs can suppress band-edge emission effectively, resulting in a remarkable decrease in the lifetime of photoelectrons in QDs from 17.1 ns to 4.9 ns. Moreover, the investigation of TTET process revealed that STSs can suppress the generation of triplet exciton and effectively inhibit band-edge emission, leading to a significant decrease in TTET from CdSe QDs to the surface acceptor. This work presented evidence for STSs influence in shaping the optoelectronic properties of QDs, making it a valuable point of reference for understanding and manipulating STSs in diverse QDs-based optoelectronic applications involving electron and energy transfer.

## 1. Introduction

Semiconductor quantum dots (QDs), also known as nanocrystals, are highly attractive materials due to their unique properties such as high molar extinction coefficient [1], high photoluminescence quantum yield (PLQY) [2,3,4,5], size- and shape-dependent optoelectronic properties [6,7], ease of solution processing [8,9], thus exhibiting a large variety of applications in light-emitting diodes [10], photocatalytic system [11,12,13,14,15], photodetectors, photovoltaic devices [16,17] and biolabeling [18,19]. However, the synthesis of colloidal QDs unavoidably results in the generation of surface trap states (STSs), which are generally considered to adversely affect some performance parameters of QDs, such as PLQY, photostability, and narrow photoluminescence (PL) spectral bandwidth, all of which are critical for optoelectronic applications [20,21]. STSs are usually produced by dangling bonds and vacancies due to the majority of atoms being located on the surface of small-size QDs [22]. These ubiquitous STSs in QDs highly likely provide pathways for nonradiative exciton recombination, thereby limiting the optical and electronic properties of QDs and being detrimental for QDs-based optoelectronic applications [23]. For instance, David R. Baker et al. reported the tuning of the emission of CdSe QDs through controlled trap enhancement [24]. Similarly, Lei Zhang et al. investigated the temperature and wavelength dependence of the energy transfer process between quantized states and surface states in CdSe QDs [25]. However, Yun Ye et al. have shown that the ultrafast hole trapping induced by STSs of QDs can be beneficial to electron transfer and inhibits the trapped hole transfer, thereby increasing the H_2_ production activity in the CdSe QDs–Ni^2+^ photocatalytic system [26]. Furthermore, recent researches exploring the impact of STSs on surface-induced energy transfer between semiconductors and molecules have revealed that STSs facilitate the energy transfer [27,28,29]. These contrasting results drawn from these studies regarding the impact of STSs highlight the importance of understanding electron and energy transfer in QDs with STSs in order to either eliminate or utilize them to enhance the performance of QDs-based optoelectronic applications.

Transient absorption spectroscopy (TAS) is an effective ultrafast “pump-probe” technique that can provide valuable information on photoinduced electron behaviors and electron transfer processes, including ground-states bleaching (GSB), stimulated emission (SE), and excited-states absorption (ESA) [30,31]. TAS has been employed in the previous studies to investigate various aspects of QDs systems, such as triplet-triplet energy transfer (TTET) from QDs to surface-anchored acceptors [32,33]. photocatalytic hydrogen evolution driven by QDs [13,26], and efficient hot carrier trapping in QDs [34]. Therefore, TAS is a feasible technique to elucidate the influence of STSs on electron and energy transfer in QDs.

In this work, we aimed to investigate the impact of STSs on electron and energy transfer in CdSe QDs and provide evidence that the photoelectrons trapped by STSs play a crucial role in photoreaction processes. Three types of colloidal CdSe QDs (referred to as ODPA-CdSe, OA/ODPA-CdSe and OA-CdSe) with similar sizes and varying degrees of STSs were synthesized using a controllable method of surface ligand chemistry. We characterized the photoelectrons transfer behaviours of ODPA-CdSe (with a few STSs) and OA-CdSe (with a lot of STSs) using time-resolved emission spectroscopy (TRES) and TAS. Moreover, the STSs influence on the interfacial TTET process between QDs as donor and surface-anchored 9-anthracenecarboxylic acid (ACA) as acceptor was explored using TAS technique with the three types of QDs. Our findings revealed that STSs can significantly decrease band-edge emission and the triplet exciton generation, consequently suppressing electron and energy transfer in QDs.

## 2. Materials and Methods

### 2.1. Materials

Cadmium oxide (CdO) and Tri-N-Octylphosphine (TOP) were purchased from Adamas-beta (Shanghai, China). Se powder, Oleic Acid (OA, AR), Methanol (AR), Absolute ethanol (AR), and Dichloromethane (AR) were purchased from SCR (Shanghai, China). 1-Octadecylamine (ODA, 97%) and 1-Octadecene (ODE, technical grade, 90%) were purchased from Alfa Aesar (Shanghai, China). Octradecylphosphonic acid (ODPA, 98.4%) was purchased from Leyan (Shanghai, China). Trioctylphosphine Oxide (TOPO, technical grade, 90%) was purchased from Sigma-Aldrich (3050 Spruce Street, Saint Louis, MO 63103, USA). 9-anthracenecarboxylic acid (ACA) was purchased from SINOPHARM (Beijing, China). Hexane (AR) and toluene (AR) were purchased from GENERAL-REAGENT (Shanghai, China). All chemicals purchased were used without further purification.

### 2.2. Synthesis Methods

A type of colloidal ODPA-CdSe QDs mainly coated with organic ligand ODPA on the surface were prepared by a chemical reaction according to references [35,36] with slight modifications. Typically, Se precursor solution was produced by adding 63.2 mg Se powder into 1.5 mL TOP with continuously stirring under the nitrogen atmosphere. Another mixture of 51.4 mg CdO powder, 269 mg ODPA, 2.5 g TOPO, and 1.5 g ODA in a 50 mL three-neck flask was evacuated at 100 °C for one hour, followed by heating to 300 °C under nitrogen flow until forming an optically clear solution. Promptly injected Se precursor into the flask upon the mixture was cooled to 280 °C, subsequently controlled the mixture temperature between 260 °C and 280 °C for QDs growth. After 8 min, stopped heating and cooled with nitrogen flow. Finally, the QDs were precipitated by adding 15 mL methanol and centrifugated at 8000 rpm for 5 min, washed 2~3 times with the mixed dichloromethane and absolute ethanol (Volume ratio = 1:1), and redispersed in n-hexane in sequence. After centrifugation, the supernatant was taken out as a stock solution.

A type of colloidal OA-CdSe QDs mainly coated with organic ligand OA on the surface were prepared by a chemical reaction according to reference [37] with slight modifications. Typically, Se precursor solution was prepared by stirring the mixture of 8 mg Se powder, 1.25 mL ODE and 45.5 mg TOP under the nitrogen atmosphere. Another mixture of 12.8 mg CdO, 565 mg OA and 4.8 mL ODE were added in a three-neck flask and evacuated at 100 °C for one hour, followed by heating to 190 °C under the nitrogen flow until forming an optically clear solution. Promptly injected the Se precursor into the flask at 190 °C and controlled the reaction temperature at 180~190 °C for 90 s, afterwards stopped heating and cooled with nitrogen flow. Finally, the QDs were precipitated by adding 12 mL absolute ethanol, and centrifugated at 8000 rpm for 5 min, washed 2~3 times with mixed dichloromethane and absolute ethanol (Volume ratio = 1:1), and redispersed in n-hexane in sequence. After centrifugation, the supernatant was taken out as a stock solution.

OA/ODPA-CdSe QDs represent a type of colloidal CdSe QDs coated with two types of organic ligand (ODPA and OA) on the surface. The QDs were prepared by a chemical reaction according to reference [32] with slight modifications. 30 mg CdO, 0.3 mL OA and 5 mL ODE were added into a three-neck flask and evacuated at 100 °C for one hour, followed by heating to 230 °C under nitrogen flow until forming an optically clear solution. Then added ODPA-CdSe (slightly less than aforementioned ODPA-CdSe in size) and controlled the temperature at 210 °C for 13 min under nitrogen atmosphere, afterwards stopped heating and cooled with nitrogen flow. Finally, the QDs were precipitated by adding 4 mL absolute ethanol and centrifugation (at 8000 rpm for 5 min), washed with mixed dichloromethane and absolute ethanol (Volume ratio = 1:1) by 2~3 times, and redispersed in toluene in sequence.

### 2.3. Treatment with 9-Anthracenecarboxylic Acid (ACA)

Surface ACA-anchored QDs were prepared by mixing ODPA-CdSe, OA/ODPA-CdSe and OA-CdSe with ACA, respectively. Added 5 mg (0.22 mmol) ACA to a concentrated solution of ODPA-CdSe in 1 mL toluene, and sonicated the mixture for 2 h at 45 °C. Then, ODPA-CdSe/ACA were precipitated by adding 1 mL anhydrous ethanol, and centrifuged at 8000 rpm for 3 min. The supernatant was removed and the residue was dispersed in toluene and stored under nitrogen in the dark at 4 °C. OA/ODPA-CdSe/ACA was prepared by the same procedure as ODPA-CdSe/ACA. For OA-CdSe/ACA preparation, 4 mg (0.18 mmol) ACA was added to a concentrated solution of OA-CdSe in 1 mL toluene, and the resulting mixture was vibrated for 10 min at room temperature. Then, OA-CdSe/ACA were precipitated by adding 1 mL absolute ethanol, and the subsequent treatment steps were same to ODPA-CdSe/ACA. The average ratio of ACA to QDs was determined by Ultraviolet-visible (UV-vis) absorption spectroscopy (Appendix A).

### 2.4. Characterization Methods

Steady state UV-vis absorption spectra were performed with a Cary 5000 UV–vis−NIR spectrophotometer (Agilent, Shanghai, China) in transmission mode. PL spectra were obtained by a HORIBA Fluorolog-3 Modular fluorescent spectrometer (HORIBA, Shanghai, China) under the excitation wavelength of 480 nm. Absolute PL quantum yields were tested by the HORIBA Fluorolog-3 Modular fluorescent spectrometer with integrating sphere. For time-resolved emission spectroscopy measurements, HORIBA Fluorolog-3 Modular fluorescent spectrometer and a Time-dependent single photon counting fluorescence lifetime test system (with a time range from 25 ps to 1 s and time resolution ~439 ps) were used under the excitation wavelength of 408 nm, and the emission wavelengths were selected from 500 nm to 850 nm at 10 nm apart. Transmission electron microscope (TEM) images were obtained by JEM-2100 Plus transmission electron microscopy (JEOL, Akishima, Japan) with an acceleration voltage of 200 kV.

X-ray photoelectron spectroscopy (XPS) was performed on a Thermo-Fisher (Shanghai, China) ESCALAB 250Xi spectrometer using Al Kα excitation, which has a 500 μm spot size in the X-ray signal acquisition area. The sensitivity and limit resolution are 0.6 eV@1,000,000 cps and 0.45 eV@100,000 cps, respectively. Samples for XPS were prepared by washing QDs with a combination of extraction and precipitation and then depositing the QDs powder onto insulated tape. A charge neutralization gun was added in the test process with the specifications of internal neutral gun current of 150 μA and external neutral gun current of 50 μA. Elemental compositions were calculated from Se 3d, and Cd 3d XPS. The binding energy was referenced to C 1s at 284.80 eV. The software Avantage (v5.9902) was used to process data and calculate the results.

Femtosecond TAS of CdSe QDs in n-hexane were carried out using a commercial femtosecond Ti: Sapphire regenerative amplifier laser system from Coherent (Astrella, America, 800 nm, 35 fs, 6.3 mJ/pulse, 1 kHz), and an automated data acquisition transient absorption spectrometer (Ultrafast Systems, Helios Fire, Sarasota, FL 34240, USA). Part of fundamental pulse (2 mJ) were transmitted to TOPAS to obtain the laser of 480 nm as pump, and the probe pulse was obtained by using 5% of amplified 800 nm output to generate a white-light continuum (420 nm~780 nm) in sapphire crystal. All experiments were performed at room temperature.

In order to obtain the desired absorbance (or concentration), we diluted the stock solution of QDs/ACA with toluene for experiment sample preparation. All samples for UV-vis absorption and PL spectra experiments were prepared in quartz cuvettes (1 cm), with optical densities of 0.3 at the excitation wavelength of 520 nm. Samples were added in quartz cuvettes (2 mm) under the argon atmosphere and the solutions were stirred continuously for TAS experiments.

## 3. Results

### 3.1. Characterization of Surface Trap States

The surface property plays an important role in the physicochemical performance of QDs. To investigate the impact of surface states on physical and charge carrier properties of QDs, two types of oil-soluble colloidal CdSe QDs capped by different surface ligands of ODPA and OA, termed as ODPA-CdSe and OA-CdSe respectively, were synthesized using organometallic hot-injection method [6,38,39]. The UV-vis absorption spectra (solid line) and PL spectra (dashed line) of ODPA-CdSe and OA-CdSe in n-hexane (Figure 1) were measured to study the photophysical impact of QDs surface states resulting from the different surface ligands. The similar peak positions of first excitonic absorption at ~517 nm for both types QDs suggested a similar size of ~2.5 nm according to size-dependent optical property [1], further confirmed by TEM results (Appendix A). The narrow band-gap PL peaks of ODPA-CdSe and OA-CdSe presented at 530 nm and 550 nm, respectively. However, in comparison to ODPA-CdSe, OA-CdSe exhibited not only a much lower band-gap PL intensity but also a distinct broad lower-energy PL peak near 730 nm, implying a large number of STSs originating from QDs surface defects [19,22,40,41]. The absolute PLQY measured by the integrating sphere technique is ~25% for ODPA-CdSe and <1% for OA-CdSe. The enormous PLQY difference between two types of QDs is consistent with the result of the PL spectra, reflecting the serious surface defects of OA-CdSe.

To determine the surface composition and stoichiometry of QDs, XPS was utilized to distinguish surface atoms from bulk atoms. Specifically, the Cd 3d and Se 3d spectra (Figure 2) were fitted with Gauss line-shape having a 30% Lorentzian character, followed by calculation of the integrated peak area [12,26,42,43,44]. The results are summarized in Table 1, while more detailed XPS peak analysis data of the Cd 3d regions and Se 3d regions are listed in Appendix A. The analysis revealed that ODPA-CdSe and OA-CdSe have surface Se atom percentages of 2% and 35%, respectively. Since the presence of dangling bond orbitals associated with surface Se is known to be related to STSs [42,43,45]. OA-CdSe, which has a higher ratio of surface Se atoms, is expected to exhibit more STSs, as observed in the PL spectra.

### 3.2. Time-Resolved Emission Spectroscopy Characterization

To reveal the influence of STSs on photoelectron behavior, time-resolved emission spectroscopy was performed on both ODPA-CdSe and OA-CdSe to study photoelectron transfer from excited state to ground state. Normalized PL kinetics for ODPA-CdSe at 530 nm and OA-CdSe at 550 nm are shown in Figure 3, while PL kinetics at 730 nm are shown in Appendix A. All PL kinetics were fitted with a multiple exponential equation:(1)$$I=C+\sum_{i=1}^{n}E_{i}\cdot exp\left[-\frac{t}{\tau_{i}}\right]$$
where C is constant, *E_i_* and *τ_i_* represent the amplitude and PL lifetime of QDs, respectively. All corresponding fitting parameters are recorded in Appendix A. It was clearly found that PL lifetime *τ*_PL_ of ODPA-CdSe (*τ*_1_ ≈ 17.1 ns at 530 nm) is longer than that of OA-CdSe (*τ*_1_ ≈ 4.9 ns at 550 nm). The faster PL decay of OA-CdSe indicated that OA-CdSe has more nonradiative pathways for electron recombination apart from radiative bandgap emission, which could be attributed to STSs trapping photoelectron. Additionally, it was also observed that PL lifetime of OA-CdSe is significantly longer at 730 nm (*τ*_1_ ≈ 37.9 ns) compared to 550 nm (*τ*_1_ ≈ 4.9 ns). As worth noting that the long-lifetime component of PL decay fitting is often traced to the participation of surface states with poor electron and hole wave functions overlap [46,47,48], it is reasonable to attribute the PL at 730 nm to STSs effect. These results indicated that STSs can trap photoelectrons and result in slower recombination of the trapped photoelectrons compared to direct recombination via bandgap emission. Combined with PL spectra, our findings suggested that OA-CdSe possesses more STSs than ODPA-CdSe, which suppress direct recombination of photoelectrons with hole in ground state.

### 3.3. Transient Absorption Spectroscopy Characterization

Real-time kinetics were tracked using TAS to gain a deeper understanding of how STSs influence photoelectron and energy transfer processes. To characterize the TAS for both ODPA-CdSe and OA-CdSe in n-hexane under argon atmosphere, a pump wavelength of 480 nm and a probe wavelength range of 420~780 nm were employed. Selected TAS were recorded between 500 fs and 5 ns, and the proposed photoelectron transfer processes of ODPA-CdSe and OA-CdSe are shown in Figure 4. Generally, TAS involves ground state bleaching (GSB), excited state absorption (ESA), and stimulated emission (SE) signals, where positive TAS signals arise from ESA and negative signals represent GSB or SE [30,31]. For the TAS of QDs, |ΔA| reaches the maximum at around 500 fs after excitation, followed by gradually decreasing with photoelectrons going back to ground state. For the negative absorption band of TAS during the time delay from 500 fs to 5 ns, the peak position of ODPA-CdSe remains almost unchanged (Figure 4a), whereas that OA-CdSe undergoes a noticeable red-shift from 526 nm to 532 nm (Figure 4b). The corresponding kinetics at different probe wavelengths are shown in Appendix A, and the TAS decay kinetics were fitted by the Equation (2).
(2)$$∆A=\sum_{i=1}^{n}A_{i}\cdot exp\left[-\frac{t}{\tau_{i}}\right]$$
where *A_i_* and *τ_i_* represent amplitude and kinetic lifetime, respectively. *n* = 3 is used for both QDs. According to the PL kinetics fitting results depicted in Figure 3, *τ*_3_ is fixed to match *τ*_PL_, specifically, at 17.1 ns and 4.9 ns for ODPA-CdSe and OA-CdSe, respectively. The corresponding fitting parameters are listed in Appendix A. For ODPA-CdSe, *τ*_1_ keeps a consistent magnitude (7.5~8.8 ps) in different probe wavelength. *τ*_2_ decreases with increasing probe wavelength (from 1229 ps at 510 nm to 44.32 ps at 535 nm). For OA-CdSe, the increase of *τ*_1_ with increasing probe wavelength (from 2.76 ps at 500 nm to 6.10 ps at 530 nm) is due to the variance of with *τ*_2_ probe wavelength. The increase in *τ*_2_ with increasing probe wavelength (from 50.03 ps at 500 nm to 93.35 ps at 530 nm) can be attributed to the strong STSs trapping process (88.3 ps) and weak PL process (4.9 ns).

The photoelectron transfer processes of ODPA-CdSe and OA-CdSe are schematically represented in Figure 4c,d, respectively. The initial ultrafast increased TAS signal indicated the instantaneous excitation of the photoactive electrons from the ground state to higher excited state. The following photoelectron relaxation can be characterized by three decay components: the fast lifetime *τ*_1_ for the electron relaxation process from E_2_ to E_1_, the longer lifetime *τ*_2_ for STSs trapping process from E_1_ to STSs, and the longest lifetime *τ*_3_ (*τ*_PL_) for the charge recombination process from E_1_ to ground state. It’s worth noting that E_2_ corresponds to the initial excited state with an excitation photon energy of ~2.58 eV (~480 nm), while E_1_ represents the bottom of conduction band at ~2.34 eV (~530 nm) for ODPA-CdSe and ~2.25 eV (~550 nm) for OA-CdSe, and STSs-induced PL suggests the presence of a trap state at ~1.70 eV (~730 nm). These energy levels are adequate to facilitate the spontaneous relaxation and transfer for excited electrons. Besides, there is one additional process, namely, photoelectrons trapped by STSs going back to ground state. For ODPA-CdSe, a part of excited photoelectrons is trapped by STSs (*τ*_2_ ≈ 379.5 ps at 530 nm), which is much faster than charge recombination process (*τ*_PL_ ≈ 17.1 ns at 530 nm). For OA-CdSe, a part of excited photoelectrons is trapped by STSs (*τ*_2_ ≈ 88.3 ps at 550 nm), which also is much faster than charge recombination process (*τ*_PL_ ≈ 4.9 ns at 550 nm). According to the kinetic fitting results at 530 nm, it was calculated using Equation (S1) that ~9.35% of E_1_ photoelectrons are trapped by STSs in 379.5 ps in ODPA-CdSe, while in OA-CdSe (kinetic fitting results at 550 nm), this percentage increases to 33.31%, with a faster trapping time of 88.3 ps. Additionally, the photoelectrons relaxation from E_2_ state to E_1_ state (*τ*_1_ ≈ 7.8 ps for OA-CdSe at 550 nm vs. *τ*_1_ ≈ 8.8 ps for ODPA-CdSe at 530 nm) could be accelerated accordingly due to faster trapping of photoelectrons, moreover, the PL lifetime in ODPA-CdSe (*τ*_PL_ ≈ 17.1 ns at 530 nm) was much longer than that in OA-CdSe (*τ*_PL_ ≈ 4.9 ns at 550 nm), indicating the fact that the lifetime of photoelectrons in excited state can be shorten by STSs remarkably. All these results reveal that a high charge trapping occurrence in the CdSe QDs trap band inhibits the utilization of the excited electrons would lead to an efficiency reduction of the surface photocatalytic reactions.

### 3.4. Influence of STSs on Triplet-Triplet Energy Transfer

Mongin et al. utilized TAS technique to successfully observe the interfacial TTET process from CdSe QDs to ACA (triplet acceptor) and illustrated that TTET can significantly enhance excited-state lifetime of CdSe QDs [32]. In order to investigate how STSs affect the photoinduced energy transfer process between CdSe QDs and ACA, three types of CdSe QDs with varying STS quantities (ODPA-CdSe, OA/ODPA-CdSe and OA-CdSe) were prepared separately and anchored with ACA. OA/ODPA-CdSe, representing CdSe QDs coated with the mixed organic ligands of ODPA and OA, was synthesized by ligand exchange with ODPA-CdSe. The UV-vis absorption and PL spectra of CdSe QDs with and without ACA anchor are shown in Appendix A. The approximate order of STSs amounts for the three types of QDs (OA-CdSe > OA/ODPA-CdSe > ODPA-CdSe) was suggested by the relative intensity of the broad lower-energy PL peak at ~730 nm in contrast to the higher-energy PL peak at ~550 nm. The ratio of ACA to QDs (ODPA-CdSe, OA/ODPA-CdSe and OA-CdSe) were determined to be 29.5, 29.1, and 27.5 respectively, by UV-vis absorption spectra (Appendix A). This is indicated that each type of QDs was anchored with similar amounts of ACA, suggesting reliability in investigating the influence of STSs for different types of QDs on electron and energy transfer between QDs and ACA. PL spectra (Appendix A) displayed noticeable PL quenching of QDs after ACA anchored, implying the occurrence of TTET and electron transfer between QDs and ACA.

To clearly monitor the electron and energy transfer and carrier population redistribution processes in the QDs/ACA system, TAS measurements at various time delays upon 480 nm Laser excitation were carried out. However, due to the influence of scattered light from pump Laser, the intersystem crossing process from singlet to triplet states, which is located in the wavelengths centred at 433 nm, could not be observed in this study. Toluene was chosen as the solvent for dissolving all samples due to solubility constraints. As illustrated in Figure 5a–c and Appendix A, a broad negative absorption band was observed for all the samples, mainly due to the GSB and SE process. The TAS of the three types of QDs before and after ACA anchoring showed quite consistent spectra within the detection wavelength window, except that the intensity decayed faster after ACA anchoring. According to research of Mongin et al., the presence of triplet state of ACA correlated to the ground-state recovery of CdSe QDs [32]. In order to study STSs influence on TTET and electron transfer, the decay kinetics of QDs and QDs/ACA at 540 nm were fitted using a stretched exponential function for TTET study [32,49], as shown in Figure 5d–f. The fitting and calculating methods are described in the Appendix A, and the corresponding fitting parameters are listed in Appendix A. It was found that the rate constants of all three types of QDs increased by about 1~2 orders of magnitude (e.g., from 7.1 × 10^6^ s^−1^ to 2.1 × 10^8^ s^−1^ for OA/ODPA-CdSe) after ACA were anchored onto QDs. The noticeable rate increasing for QDs after coating with ACA acceptor is attributed to the TTET process at the interface and electron transfer between QDs and ACA [32]. A detailed analysis and explanation are as follows.

The rate constants <k> of pristine QDs at 540 nm (Figure 6a) follow the order of ODPA-CdSe (1.01 × 10^8^ s^−1^) > OA/ODPA-CdSe (7.10 × 10^6^ s^−1^) ≈ OA-CdSe (7.57 × 10^6^ s^−1^), indicating that STSs can retard the rate of electrons going back to ground state by SE process. This is in consistent with the study of STSs influence on electron transfer in the Section 3.3.

The rate constants for ODPA-CdSe, OA/ODPA-CdSe and OA-CdSe increased after ACA was anchored. According the previous literature [32], the rate constant difference between QDs with and without ACA was defined as <k>_TTET_ (the rate constant of TTET). To demonstrate that STSs is a significant influencing factor in the TTET process, the <k>_TTET_ of the three types of QDs at 540 nm were calculated and shown in Figure 6a,b, respectively. <k>_TTET_ of QDs follows the order of ODPA-CdSe (2.55 × 10^9^ s^−1^) > OA/ODPA-CdSe (2.01 × 10^8^ s^−1^) > OA-CdSe (7.47 × 10^7^ s^−1^), which indicates that the rate of SE process is inversely proportional to STSs number in QDs with the presence of ACA. Additionally, change of triplet energy transfer rate constants <k>_TTET_ from QDs to ACA at 540 nm with STSs percentage in QDs is shown in Figure 6c.

As shown in Figure 1, there are six processes happened in QDs with ACA presence after photoexcited: (1) electrons are excited to E_2_ state from ground state, (2) electrons relaxation from E_2_ state to E_1_ state, (3) electrons go back to ground state from E_1_ state (SE process), (4) electrons trapped by STSs, (5) electrons go back to ground state from trap state, (6) electrons in E_1_ state generate triplet excitons. There are three competing processes directly involved in QDs/ACA system: (3) SE process, (4) STSs trapping, and (6) triplet exciton generation. As illustrated in Appendix A, the PL quenching observed after anchoring ACA onto QDs indicates that when TTET occurs, the SE process is greatly suppressed. Compared to ODPA-CdSe, the presence of STSs in OA/ODPA-CdSe and OA-CdSe has the potential to decrease the lifetime of excited state electrons, effectively suppressing SE process and triplet exciton generation through STSs trapping. This is highly disadvantageous in triplet exciton transfer to ACA. Therefore, STSs in QDs play a negative role in the generation and transfer of triplet exciton.

## 4. Conclusions

In summary, we investigated the impact of STSs on electron transfer in CdSe QDs as well as TTET from CdSe to ACA anchor using the femtosecond transient absorption spectroscopy. Three types of CdSe QDs (ODPA-CdSe, OA/ODPA-CdSe and OA-CdSe) with similar sizes but various amounts of STSs were synthesized by controlling surface ligands and reaction conditions. The PL spectra indicated that STSs amounts in the three types of QDs follows the order of OA-CdSe > OA/ODPA-CdSe > ODPA-CdSe. TRES and TAS were performed to study the influence of STSs on photoelectrons transfer processes. TRES revealed that STSs can trap photoelectrons, leading to a decrease in PL intensity and shorten the lifetime of photoelectron from 17.1 ns (ODPA-CdSe) to 4.9 ns (OA-CdSe). Similarly, TAS study indicated that an increased presence of STSs can decrease the lifetime of trapping process from 379.5 ps (ODPA-CdSe) to 88.3 ps (OA-CdSe), and increase proportion of trapping electrons from 9.35% to 33.13%. Furthermore, the influence of STSs on interfacial TTET from QDs to surface-anchored ACA was studied by TAS. By comparing the TAS of ODPA-CdSe, OA/ODPA-CdSe and OA-CdSe before and after ACA anchored, three competing processes related to TTET efficiency were identified, including SE process, STSs trapping, and triplet exciton generation. The TAS results suggested that the STSs amounts in CdSe QDs can greatly suppress TTET by decreasing triplet exciton generation, i.e., the more STSs, the smaller TTET efficiency. This work provided a deeper understanding of how STSs affect electron and energy transfer processes. The discoveries and research methodologies hold promise for broader applications in Cd-free or other “green-chemistry” QDs, such as InAgSe, InP, CuInS, etc. We anticipate that this research will serve as a valuable reference for the advancement of optoelectronic applications based on QDs, extending beyond CdSe.

## Data Availability

Data are available on demand.

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
