# Peer review of "Impact of Surface Trap States on Electron and Energy Transfer in CdSe Quantum Dots Studied by Femtosecond Transient Absorption Spectroscopy"

_nanomaterials, 2023, doi:10.3390/nano14010034_

Round 1

Reviewer 1 Report

Comments and Suggestions for Authors

Nice paper, I only have a small comment: line 350 ... were calculated and shown in Figure 6 (b) and (d), respectively. Figure 6 only has parts (a) and (b) not (d).

Reviewer 2 Report

Comments and Suggestions for Authors

The manuscript is devoted to the study of the dynamics of photoexcitations in CdSe NCs capped with different ligands in the presence/absence of a molecular acceptor by means of transient absorption (TA) spectroscopy.

The authors claim that the surface trap states (STS) suppress photoluminescence (PL) and induce the reduction of PL lifetimes. In fact, these observations were made numerous times last decades. Next, the authors study TA for the QDs anchored to molecular acceptors, similar to that reported in Ref.27. Interestingly, the data from this subsection and their analysis do not correspond to that presented in a previous subsection. No direct evidence for triplet-triplet energy transfer for all three cases was presented. Other mechanisms were not considered. The real conclusion that can be made here is that the calculated rate of assumed energy transfer correlates with the assumed density of trap states. This is far behind the claim of “it a valuable point of reference for understanding and manipulating STSs”. I believe that the manuscript suffers from the lack of novelty and deep analysis, and thus should not be considered for publication. 

Reviewer 3 Report

Comments and Suggestions for Authors

H. Dou et al. report about Femtosecond transient absporption spectroscopy to give insights into the impact of surface trap states on electron and energy transfer in CdSe quantum dots

The methodology appears well prepared and well described and both scientifically and technically mature.

The major shortcoming I see with the paper is that it is to look considerably outside the box what has been done on the field using other scientific and technical approaches. A few hints are given in the following:

1_ As only one technique was employed, the title of the paper should be de-armed (become more modest), e.g. reduced to "Impact of Surface Trap States on Electron and Energy Transfer in CdSe Quantum Dots Studied by Femtosecond Transient Absorption Spectroscopy"

2_ Source [22] Introduction (line 41 ff): It claims STSs for UV irradiation of small QDs. I expect the authors to review previous work that claims STS and STS-induced energy transfer as ground for optical effects for a variety of QDs also in the visible range.

3_ The outlook "This work ... provide(s) powerful evidence for optimizing QD-based optoelectronic applications through STS manipulation" in the Conclusions (line 387 ff.) should be mirrored also already in the introduction with respect existing and previous QD engineering,

a) e.g. existing works on crystal facets in colloidal QDs

b) on the synthetic control of such facets

c) on surface-induced energy transfer investigated (back in 2007 and 2012)  

d) as well as non-FRET energy transfer in QD surfaces induced by "disturbing" molecules,

4_ To address future applications, as the paper only discusses CdSe quantum dots, it should also look into Cd-free or other “green-chemistry” related systems (e.g. InAgSe) and sketch what part of the results can be employed in that sense.

Two technical issues are to be addressed:  

5_ line 236: in Figure 3, I expect an Instrumental Response Function (i.e. the time development of the excitation function)  

6_ line 343 ff: Figures 6 a) and b) I would expect the bars (blue, yellow; green) to carry error bars.

#

Reviewer 4 Report

Comments and Suggestions for Authors

The authors perform multiple characterization techniques (TEM, XPS, PL, absorption, time-resolved PL, and transient absorption spectroscopy (TAS)) to study the influence of surface trap state (STS) on the CdSe QD emission. In particular, by varying the STS density and including surface-anchored triplet acceptor (in this case, ACA), they experimentally show that the STS inhibits triplet-triplet energy transfer (TTET).

Overall, the reported results are reasonable and of interest to the community studying CdSe QD emission. The draft was written quite clearly. Therefore, I recommend publication in Nanomaterials, given that the authors can address the comments below:

1.      The authors should give a more thorough literature review. In particular, there are other reports studying the relation between STS and TTET (for example, Bender, Jon A., et al. J. Am. Chem. Soc. 140.24 (2018): 7543-7553; Luo, Xiao, et al. Nat. Commun. 11.1 (2020): 28), with some studies reported that the STS can facilitate TTET, instead of inhibiting it. The authors should compare their proposed mechanism with the other existing mechanisms.

2.      In some sentences, the authors mentioned the TTET and electron transfer (between STS and ACA) together. However, these two processes are different. TTET is energy transfer (i.e., the electron and hole are both transferred), while electron transfer refers to negative carrier transfer. The authors should differentiate these two processes.

3.      The authors related the PL peak at ~730 nm to the STS. Does this mean that the STS energy is ~1.7 eV? The authors should discuss the possible energy of the trap states (Fig. 4) based on their measurement and whether this energy matched the theoretically expected STS energy level.

4.      What is the expected lifetime of the STS? Is it the tau_1 at 730 nm (71.23 ns)? Given that the decay rate in Fig. 4 is known, it is possible to get the ratio between band-edge emission and STS emission under the steady-state condition. Does this ratio match the ratio obtained from the PL measurement (Fig. 1)? If it does not, what are the possible reasons?

5.      What is the significance of the biexponential nature of the PL emission shown in Fig. 3 (also, Table S3)? Table S3 should also show the value of A_i used in Eq. (1). Additionally, I suggest changing A_i in Eq. (1) to another symbol to differentiate it from the A_i in Eq. (2), which is related to absorption.

6.      What are the possible reasons that the OA-CdSe has much more STS (in %) than the ODPA-CdSe?

7.    Based on the experimental results, the transfer rate constant seems related to the STS percentage. Hence, in addition to Fig. 6b, I suggest plotting the transfer rate constant vs STS percentage.

Round 2

Reviewer 2 Report

Comments and Suggestions for Authors

The following comments must be addressed:

1)     Why do QDs of the same size (ODPA and OA) have so different absorption spectra? Why is confinement absent in the case of OA-caped QDs?

2)     The fitting of PL kinetics in Figure S2 is wrong.

3)     The authors calculated the average decay time accurate to two decimal places (17.08 ns). However, what is the measurement error?

4) The text must be spell-checked
